# Simulation of Intellectual Property Management on Evolution Driving of Regional Economic Growth

**Xiran Yang** *  and **Yong Qi**

School of Intellectual Property, Nanjing University of Science & Technology, Nanjing 210094, China
* Correspondence: xr95@njust.edu.cn

**Abstract:** The input, application, and transformation of intellectual property can significantly promote the economic development of a region, but the path and operation mechanism of intellectual property management on regional economic growth are not very clear. System dynamics theory was used to analyze the driving force and resistance of intellectual property management from macro to micro. With the help of system dynamics theory, equations were constructed to simulate the process path and force from intellectual property management to regional economic growth, and sensitivity analysis was used to find sensitive influencing factors in the system. The following conclusions were drawn: (1) intellectual property affects regional economic growth from the macro level, such as intellectual property investment, policies, and the construction of rules and regulations; (2) enterprises, whether in the industry, universities and research institutes, or in this system, are the main body to create innovative benefits and ultimately promote regional economic growth; (3) the continuous investment of intellectual property resources and the driving force of enterprise innovation are all sensitive factors of this system. The government should give full play to its functions and strengthen the management of intellectual property in order to enable the regional economy to obtain high quality development. Through the study of the cooperation between the subjects in the process of intellectual property management activities, the integration and allocation process of factors and resources has been researched, and revealing the action process and dynamic change process of technological innovation activities on economic growth also have been revealed.

**Keywords:** system dynamics; intellectual property management; regional economic growth





## 1. Introduction

Under the new situation, the intellectual property strategy, as an important national development strategy, serves the goal of building an innovative country. With the promotion of the strategy of building an innovation-oriented country, the central and local governments have taken improving the management level of intellectual property rights as an important means to drive regional economic development. The essence of an intellectual property strategy is to promote and protect innovation. Enterprises continue to carry out technological innovation, produce new products, and gain innovation benefits, and the regional economic level has also achieved rapid development [1]. At present, the speed of economic development is slowing down in our country. In order to achieve breakthroughs and development, we must vigorously develop technological innovation, promote industrial upgrading, and optimize and adjust the industrial structure. Only by improving the management level of intellectual property rights can the country improve the technological innovation ability of enterprises. Enterprises are the main body of application of intellectual property rights, and intellectual property management is an important link in the operation and management activities of enterprises. Intellectual property management departments play an important role in the operation and management of enterprises, from the positioning of the overall management system to the setting, staffing, and actual functions of the

management departments. The economic benefits of intellectual property can be improved through intellectual property management.

The influence of intellectual property management activities on regional economic growth can be clearly measured by means of production function. However, how does an intellectual property management system function in the whole process of economic growth? Is it directly, or does management activity function through intermediary activities? How does the multi-subject function in the whole management activity? How does the service platform built by the government in the management activities serve and interact with enterprises? This study combined the intellectual property management system with the regional economic system, used the system dynamics to study the collaboration and innovation activities among multiple subjects, and revealed the driving path and effect strength of intellectual property management activities on regional economic growth. Firstly, the relevant literature on the drive of intellectual property rights to economic growth was reviewed, and it was found that the existing research mainly focuses on the econometric research. In order to explore the interaction between subjects and management activities, the system dynamics method was introduced to analyze the interaction path between subjects and activities, and the system dynamics flow diagram was constructed. Then, the system dynamics model was constructed, and Jiangsu Province, whose intellectual property management level ranks the second in China, was selected as the research sample. The internal driving path and factors of intellectual property management activities on economic growth were studied and simulated, and a sensitivity analysis was carried out.

The path of how intellectual property management activities promote economic growth is not clear. The simulation of system dynamics can help us figure out the key paths and factors that intellectual property management activities drive economic growth, to make the enterprise and the government better strengthen the intellectual property management level, to encourage enterprises to produce more high-value patents, and to promote their transformation to become the driving force for industrial chain upgrading, so as to promote regional economic growth [2]. It also plays a role in benchmarking and leading the intellectual property management activities of other provinces.

## 2. Literature Review and Theoretical Basis

### 2.1. Literature Review

The domestic and foreign scholars have conducted a lot of research on the impact of intellectual property on regional economy. In this aspect, the research mainly focuses on several aspects: (1) The important role of intellectual property resources. The distribution of intellectual property resources is uneven in regions, and the eastern region obviously gathers intellectual property resources, forming an innovation gathering highland in the resource agglomeration area [3]. Intellectual property is intangible assets. Under the guidance of an intellectual property strategy, enterprises can effectively utilize this scarce resource and turn it into a competitive advantage [4]. In these studies, intellectual property resources mostly refer to the broad sense of knowledge resources, including human, financial, material, and other resources related to intellectual property work. In the narrow sense, intellectual property refers to copyrights, patents, trademarks, new plant varieties, integrated circuit layout designs, geographical indications, etc. [5]. Under the background of digitalization, the use of intellectual property resources is faced with uncertainty, heterogeneity of resources, non-formalization of capabilities, etc. [6] In the process of innovation, search, extraction, and selection mechanisms should be introduced to select suitable resources [7]. Digital intellectual property has permeated the economy and is increasingly becoming a means of enhancing competitiveness [8]. (2) The impact of the intellectual property strategy and policy on innovation performance or regional economy. As Bryan found, intellectual property policies and rules are helpful to promote economic growth in developed countries. In developing countries, the role of intellectual property protection in promoting economic growth is still controversial [9]. The ultimate purpose of implementing an intellectual property strategy is to promote innovation and the creation and application

of knowledge achievements. The bootstrap-SEM (structural equation model) model is also used to measure the transformation path of intellectual property. It was proven that the policies and R&D (research and development) investment have significant indirect positive effect on economic benefit, and the effect is the largest [10]. In future development, green intellectual property as a strategic resource will be valued in sustainable development, and the concept of sustainable development will be incorporated into the organization's development plan in the innovation management activities of intellectual property [11]. Through studying the system, policy and practical experience of foreign standardization organizations and standardization enterprises, the standards of industrial alliance and intellectual property policy for protecting international trade have formed [12]. The key is to study the correlation between a quality policy and an intellectual property policy, as well as the strength determined by innovative enterprises [13]. (3) Research on measuring the contribution of intellectual property to the economy. This kind of research uses different research methods to quantitatively study the contribution or role of intellectual property to economic growth. For example, the contribution of intellectual property to economic development is measured by using the production function containing intellectual property rights [14]. Using the entropy value method [15], or using cointegration and error correction model or panel data to measure the relationship between the two [16], the relationship between the two has been proven quantitatively. (4) The promoting effect of intellectual property protection on economic growth. Some scholars found through empirical studies that, when the technological level of developing countries is far from that of the world's advanced technology, strengthening intellectual property protection will inhibit imitation activities, which is not conducive to technological progress and economic growth [17]. Similar to this view, some scholars empirically found that, for the economically underdeveloped regions, the adoption of adaptive IPR (Intellectual Property Rights) protection policies and the optimal IPR protection intensity can promote economic growth [18]. There are also empirical studies by some scholars who believe that, for regions with high R&D investment intensity through independent R&D channels and high FDI intensity through international technology transfer, intellectual property protection plays a greater role in promoting economic growth [19]. The study found that intellectual property protection has a positive effect on innovation sustainability. This is an official means of protecting innovation, and intellectual property rights help companies continue to innovate. In addition, these results support the effectiveness of the intellectual property regime [20]. The protection of intellectual property is a mechanism of economic appropriation that rewards innovative efforts [21].

In all the above studies, the influence on technological innovation is discussed without exception, indicating that technological innovation plays an intermediary and bridge role in the management process of intellectual property rights. In the research of scholars, many researchers directly study the relationship between intellectual property and technological innovation or study technological progress or innovation ability, or they study from a single perspective, such as the impact of intellectual property protection on technological innovation [22]. Therefore, in this study, the enterprise's technological innovation was considered as an important variable and an important factor in the system.

A consensus was reached on the role of intellectual property management activities in promoting economic growth. Previous studies have been conducted from the following aspects: (1) the impact of intellectual property input (according to the principle of the relationship between input and output, the larger input will obtain the greater benefit); (2) empirically obtain the value contribution of intellectual property management activities to economic growth by means of measurement; (3) the government's role in the economy. The economic and management functions of the government should be effectively played in intellectual property management activities, including providing public services and financial investment, promoting industry–university-research cooperation, and protecting intellectual property rights. The above research broke through the simple value measurement or the influence of a single factor or activity and focused on the cooperation between

government, industry, university, and research subjects in intellectual property management activities, technological innovation of enterprises, the formation of innovation effect, and the path of promoting economic development.

*2.2. Theoretical Basis*

From the existing research literature, there are three main lines of research in this field. The first line is the resource main line. Resource input is directly proportional to R&D output. Resources here generally refer to policy resources, strategic resources, human resources, and intellectual property resources. The inherent advantage of resources is not only bring profits to enterprises, but also influence regional economic growth accordingly. The second line of research is multi-agent collaboration. The implementation of intellectual property rights is a national strategy. In order to build an innovative country, it is neither the responsibility of the government itself, nor the realization of enterprises or a single subject. It requires the coordination of multiple subjects such as the government, enterprises, and research institutes. In this process, not only the financial support of the government is needed, but resources and reasonable allocation are also needed to promote the maximum wealth creation of intellectual property. The government focuses on the establishment of intellectual property operation platforms and intermediary institutions, through which the financing of intellectual property pledge and financing and intellectual property transactions can be realized, so as to achieve the optimal allocation of social resources and improve the efficiency of intellectual property utilization. The third line of research is the mediating effect of technological innovation. Intellectual property itself is the resources of society and enterprises. Only through transformation activities, using intellectual property to carry out technological innovation activities, developing new products, and selling them on the market, can the sales revenue of new products be realized. Intellectual property rights bring new and more value-added activities to enterprises through their technical activities than the original products.

The drive of intellectual property rights to the regional economy cannot be realized by a simple activity. Intellectual property resources, namely human and property, must be invested; then, through the cooperation of the government, industry, universities, and research institutes, research and development activities should be carried out through cooperation between enterprises and scientific research institutes. In this process, the investment of R&D funds is supported by local government financial funds and is invested by enterprises themselves in operation. The government plays multiple roles in the process of achieving economic growth. The first is the role of financial backer. In the construction of innovative provinces, local governments attach importance to innovation, respect intellectual property rights, and invest financial funds to support the development of intellectual property rights. Another role of the government is that of a service-oriented government. The government provides a variety of public services for the strategy of a province with strong intellectual property rights, such as establishing intermediary platforms and intellectual property trading markets, or building bridges between enterprises, schools, and research institutes to promote cooperative R&D. The third role of the government is to provide corresponding policy incentives and sound institutional guarantees to promote the healthy development of intellectual property rights. In the whole activity, the government should also carry out the whole chain protection of intellectual property, especially in the cooperation and transaction of intellectual property with international countries, where intellectual property protection becomes more important. In this process, enterprises make full use of government policies and financial support, supporting corresponding R&D funds, cooperating with scientific research institutes or independently researching and developing, or purchasing needed patents through intellectual property trading platforms to carry out technological transformation and innovation, then produce innovative products. The product is traded in the market to obtain new product revenue. In the process of value increment, intellectual property management activities run through the whole process, including the input, application, transformation, and protection of intellectual property. In

this process, multiple disciplines are harmoniously cooperated, various resource elements are integrated, and knowledge is used and transformed.

Through the above analysis, the system of this value chain can be divided into three subsystems. The first is the government support system, in which the government invests financial funds, establishes intermediaries and intellectual property trading centers, promulgates incentive policies, and improves the legal system to protect intellectual property rights. Secondly, there is the technological innovation transformation system, in which enterprise innovation includes three aspects: they can independently develop and carry out original innovation activities; they can cooperate with schools and research institutes to carry out technological innovation; they can also carry out intellectual property pledge financing, intellectual property transfer, and licensing through the intellectual property trading center, so as to obtain financial support, productize patents as soon as possible, and obtain profits. In the system with these two subjects as the center, a complete intellectual property input, application, transformation, and protection activities are formed, constituting the intellectual property management.

As the main bodies take on different roles and responsibilities, the relationship between each factor depends on the game between different subjects and the cooperation abilities. The government plays an important role in the system, the inherent "night watchman" role and the subject undertaker of service-oriented government. The government is not only the policy of providers and financial spenders, but it also plays a good role in serving enterprises, universities, research institutes, and other major institutions, and it adjusts the allocation of resources. The activities between these systems are both synchronous and transitive. In the process of connection between subjects, there are causal feedback and structural changes among these elements, and they are constantly evolving.

With the help of system dynamics (SD), it can help to understand the complex relationship between the activities of these subjects and the feedback relationship between the elements. By studying the cooperation relationship between subjects and the driving relationship between factors, we analyzed the causal feedback and directionality of the system, came to understand the close dependence relationship between subjects, analyzed the power and resistance, and explored the purpose of the system under changing internal and external conditions. The adaptability of this method was analyzed as follows: (1) It took a process from intellectual property management activities to regional economic growth. In this process, the impact of cooperative activities of multiple subjects on the results was uncertain. How to strengthen the cooperation between the main body and to promote the optimal allocation of resources, different management modes, and cooperation methods will bring different results. (2) The realization of value added was to generate innovative products and realize the value increment of new products through the technological innovation activities of enterprise subjects. In the transformation of intellectual property rights, enterprises have the autonomy and initiative to choose self-creation, purchase, or cooperative research and development, which will have different effects on regional economic growth. The role of government is important. (3) Value realization has to go through a long process of input, application, and transformation, and there is a time lag in this process. The time difference depends on the length of time from intellectual property rights to technological innovation, the output of innovative products, value realization, and so on. (4) Through the establishment of the causal feedback loop, the direction and process of interaction between elements can be deeply understood, the structure of each element and the structural reasons of the elements can be deeply analyzed, and the subject behavior and decision-making behavior can be explained.

## 3. System Feedback Mechanism Analysis and Model Construction

### 3.1. System Evolution Process Analysis and Feedback Loop Generation

3.1.1. Analysis of Evolution Driving Process of Intellectual Property Management System

Government is the main body of administration, undertaking administrative and economic functions. For the sake of local economic development, the government has the

decision-making power to use fiscal revenue to invest and support the development of intellectual property rights. In the implementation of innovation strategy, local governments take innovation as the driver of the local economy, issue various incentive policies and measures, improve various rules and regulations, and regard the development of intellectual property rights as a strategic resource and core element to enhance regional competitiveness and enhance the local economy. From the perspective of economics, input and output are in direct proportion. In addition, it has been proven by scholars that more input in intellectual property can have a positive impact on the economy. The intensity of financial input depends on the economic foundation of a region and the degree of emphasis on intellectual property management. Financial expenditure is spent on the input of intellectual property factors. The administrative function of modern government reflects more service government function. In addition to financial support, local governments build public service intermediary platforms and intellectual property trading platforms, provide information services and intellectual property pledge financing, and help enterprises to transfer patents and technologies. We established open innovation platforms, digital resources, and application scenarios for enterprises; provided technology verification and demonstration opportunities for micro, small, and medium-sized enterprises; drove upstream and downstream enterprises to accelerate digital transformation; and improved the digitalization level of the whole process and chain. We improved mechanisms for mining, cultivating, and supporting high-growth technology-based micro, small, and medium-sized enterprises and strengthened the development of incubators, makerspaces, and other carriers of scientific and technological entrepreneurship. The government issued precise policy support to guide leading enterprises to increase R&D investment and build high-level R&D institutions. The leading enterprises guide the universities and institutions to build collaborative innovation consortium of all innovation subjects.

As a micro-subject, modern enterprises also realize that intellectual property is an important channel for enterprises to upgrade and gain competitiveness. They establish intellectual property strategy, win financial support from the government, attach importance to technological innovation of enterprises, apply for various innovative projects, promote patent transformation, and form innovative products. No matter in any process of the input, application or transformation of intellectual property rights, enterprises will encourage the development of intellectual property rights under the guidance of the strategy, apply for patent projects, apply intellectual property rights to the new products, and then trade to obtain innovation benefits.

The above analysis forms the following causal feedback loop:

I (Regional GDP—government financial input—intellectual property management input—knowledge achievement stock—technology market transaction amount—transformation rate of scientific and technological achievements—new product sales revenue—innovation revenue—industrial output—GDP growth rate—regional total output value).

The causal feedback path reflects the government economic functions and service-oriented government functions. Government attaches great importance to intellectual property contribution to the economy, increasing financial investment and building the intellectual property rights trading center and service platform, to promote intellectual property pledge financing, etc., and it helps enterprises to carry out technical innovation and form innovation benefits.

II (intellectual property strategy—policy support—intellectual property pledge financing capacity—market purchasing power—new product market demand—innovation impetus—enterprise innovation factor input—new product—new product sales revenue—innovation income).

This feedback path reflects the political function played by the government, which utilizes policy, institution, and other functions to formulate intellectual property strategy and lead enterprises to innovate.

### 3.1.2. The Driving Mechanism of Technological Innovation on Performance

There are several ways for enterprises to innovate. First, they should rely on their own strength to conduct research and development. Enterprises obtain certain financial support from the government and rely on their own reserve of scientific research personnel, research and development, and technological innovation activities to form new products. Second, there is a cooperation in research, development, and innovation. With the help of the government, enterprises cooperate with research institutes and schools for joint research and development and promote the collaboration among enterprises, universities, and the government. Knowledge is transformed and promoted in the collaboration process, and a knowledge exchange occurs. The cooperation between the two sides reflects win–win cooperation and complementary advantages. In industry–university–research cooperation, enterprises play a leading role in technological innovation, leading the innovation process and playing a leading role in the process of putting forward innovative ideas, innovating products, investing in technology research and development, and the industrialization of achievements. Third, the IP curve is used to overtake. Patent ownership is purchased or used right through the intellectual property trading market, valuable technical solutions were obtained, important technical support for enterprises or technology development were provided, and research and development time were shortened. Enterprises actively engage in research and development work, enhance the cooperation, and exchange with the government and scientific research institutes.

Market competition prompts enterprises to continuously explore new paths for technological innovation activities. Firstly, they should carry out effective allocation and reasonable adaptation of innovation resources. Secondly, they should obtain patents from the above ways, use patented technologies to promote innovation, develop innovative products, and obtain benefits of innovative products in market transactions. In the process of innovation, the integration of innovation risk and market risk enhances the uncertainty of enterprise innovation.

(Driving force of technological innovation—industry-university-research—R&D input—new product revenue).

This cause-and-effect feedback mechanism mainly reflects the path that obtains the enterprise main body innovation income. In this process, enterprises, universities, and research institutes cooperate to share innovation results and generate innovation benefits.

In the same industry, market competition promotes imitation and innovation among enterprises. In the industrial chain, upstream enterprises carry out innovation, and downstream enterprises avoid being eliminated by upstream enterprises. They constantly follow the footsteps of upstream enterprises, imitate, follow up, and form innovative technologies and products. From the perspective of the whole market, enterprises in the industrial chain, driven by innovation, cooperate with upstream and downstream enterprises in continuous technological innovation. In the process of innovation, the enterprise keeps exploring and even obtains the technological achievements in the previous new fields of the industrial chain. In such a system of constantly pursuing technological innovation, the innovation ability of enterprises in the whole industry and industrial chain is improved as a whole. The innovation performance has improved accordingly.

### 3.1.3. Subsystem Function Analysis

From the above analysis, it can be seen that, with the support of government financial funds, enterprises formulate intellectual property strategies, increase R&D investment under strategy and development, cooperate with industry–university–research institutes, or rely on their own advantages, and they increase the patent ownership of core technologies, which is the output process of intellectual property. In order to obtain profits, enterprises promote the industrialization of patented technology, and realize profits in the market; then, the market value of intellectual property rights can be obtained. In this process, the alliance of government, industry, university, and research and the appreciation of intellectual property are realized, which is the inlay and interaction of the intellectual

property management process and the enterprise technological innovation in the process of realizing the value. This process is a parallel and constantly interwoven process and a process of intellectual property to promote enterprise technological innovation. In this process, the interaction and collaboration between multiple subjects are also realized.

The path of intellectual property management activities to promote regional economic growth is an evolutionary process in which multi-agents cooperate and fulfill their respective functions in different fields. This process is influenced and acted upon by the external environment. The market competition will promote the cooperation between the main body and, at the same time, will make the main body in order to achieve their own mission to actively promote various activities. In this case, the whole system can be divided into several subsystems: an intellectual property management system, a technology innovation driven system, and a government policy support driven system. The government policy support system is the guaranteed system of the large system. The function of this subsystem is derived from the important position and role of the main body. The government first is the perception of the environment. The fierce market competition has made governments realize that innovation can win a head start in the future development of science, technology, and industry. How to create a sound scientific and technological innovation ecosystem to drive high-end industries to lead has become the main task of governments. With the fierce development of innovation competition, the industrial innovation ecosystem that affects the application effect of scientific and technological innovation has also become an important aspect of reform and construction; thus, the strategic change has begun. In this system, under the guidance of the government's innovation strategy, enterprises, research institutes, and other subjects are incorporated into this system. The other is enterprises' intellectual property management system. In the ecological pattern of division of labor and cooperation in innovation, these innovation entities, such as research institutes, universities and small and medium-sized enterprises (smes), undertake their respective main tasks. At present, the main bodies of innovation have undergone significant changes. A large number of innovative large technology enterprises have emerged due to their ability to adapt to market changes, and are increasingly becoming the leader and main force of scientific and technological innovation and industrial innovation. With the emphasis on technology research and development and the ownership of core technologies as the main driving force, the company efficiently integrates internal and external resources and makes a large amount of investment research and breakthrough innovation in scientific and technological innovation, industrial innovation, and even basic application research. Enterprises with a strong open innovation and continuous innovation ability, a strong innovation culture atmosphere in the whole organization, an efficient integration of internal and external resources, and a strong radiation effect of innovation results are the main force of current and future innovation, and they are also the implementer of innovation-driven development.

The intellectual property management system is interwoven in the activities of the government support system and the enterprise innovation system, such as the input and integration of intellectual property resources, the transfer and transformation of patents, and so on. With the participation and cooperation of the main body, economic growth is realized. The operation of the whole system has been completed.

*3.2. Generation of System Dynamics Flow Diagram*

In this study, the system dynamics software Vensim was used to draw the dynamic flow diagram (as shown in Figure 1), and then variables were defined and assigned to carry out the dynamic simulation and the simulation of regional economic growth through intellectual property rights.

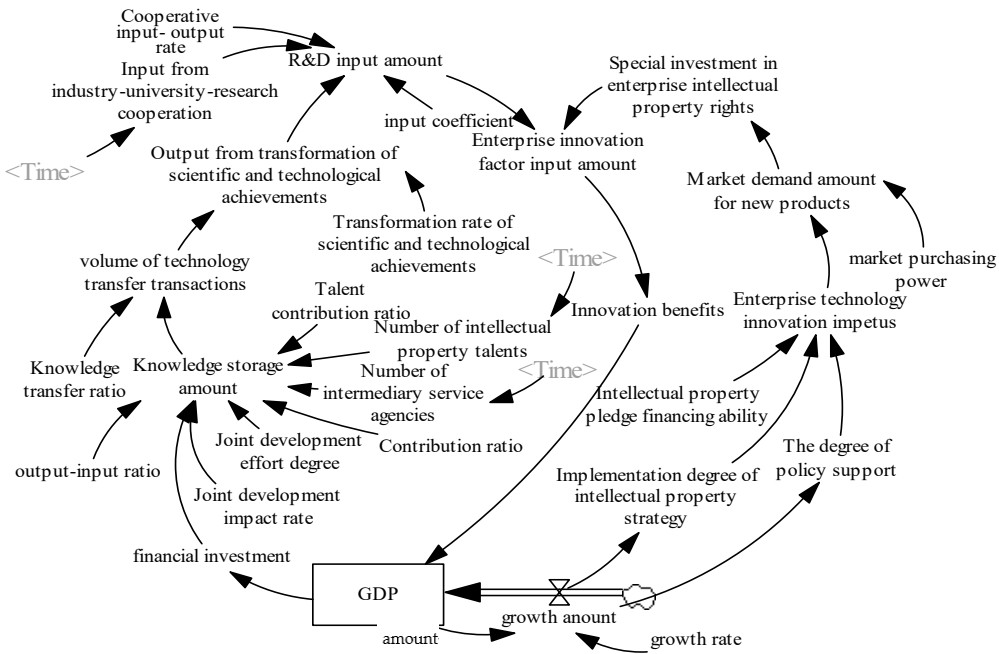

**Figure 1.** System dynamic flow of IPR influence on regional economic development.

## 4. System Dynamics Simulation of Driving Mechanism of Intellectual Property Management to Regional Economic Development

### 4.1. Basic Assumptions of the Model

The increase of intellectual property investment intensity can promote regional economic growth. Financial investment and scientific research personnel and intermediary service agencies, to promote the increase of the number of patent applications and grants and other intellectual property rights, rely on the government, enterprises, research institutes, institutions of higher learning, and cooperation between output, and then, in the market for technology transfer, the enterprises put technology into products, form innovation, and promote regional economic growth.

The government establishing an intermediary service center to provide public services, promoting technology transfer and intellectual property rights transaction activity, for enterprise, government, and research institute cooperation to strengthen the enterprise technology innovation activities and promote the enterprise's own economic growth. In this pattern, multiple subjects can play their role in the system and jointly contribute to the harmonious operation of the system.

The innovation strategy promotes the implementation of the regional intellectual property strategy. The government has issued relevant policies to support the implementation of the strategy and carried out innovative activities, such as intellectual property pledge financing, to solve the shortage of an enterprise innovation fund. The government should exert its economic functions to stimulate the market demand for new products, improve the market purchasing power of new products, promote enterprises to generate a technological innovation impetus, increase the input of enterprises in innovation factors, and produce more new products to realize innovation benefits and promote regional economic growth.

### 4.2. Parameter Estimation and Variable Model Construction

Since the implementation of the intellectual property strategy in Jiangsu Province, intellectual property has developed rapidly and made great contributions to the economy. This study selected the intellectual property data of Jiangsu Province to simulate the evolution and promotion process of intellectual property management on regional economic growth in Jiangsu Province from 2015 to 2020. Intellectual property data were derived from the Jiangsu Statistical Yearbook, China Statistical Yearbook of Science and Technology, etc.

The relationship between the main variables of the model is explained as follows:

The data of variables can be obtained in three ways: (1) directly from the statistical yearbook, such as the initial value of regional gross output value, the number of intellectual property talents, the number of intermediary service agencies, and the input of industry–university–research cooperation; (2) using regression analysis, such as knowledge stock, R&D investment, etc.; (3) using the acquisition of expert consultation law, such as the intensity of joint development, the implementation of intellectual property strategy, and the intensity of policy support, which will be valued as 1 and given weight.

Here are the main equations of variables:

(1) GDP growth = INTEG (growth rate, initial value of regional gross output);
(2) Initial value of regional gross output value = 70,116.4, growth rate = 0.0184;
(3) Knowledge stock = Number of intermediary service agencies * Contribution of intermediary service agencies * 0.25+ Number of intellectual property talents * contribution of intellectual property talents * 0.25 *+ Joint development efforts * Joint development impact factor * 0.25+ Financial input * financial input-output factor * 0.25 + 72.38;

(Knowledge stock includes statistics of intellectual property resources in a broad sense, such as patent application and authorization, intellectual property related talents, intermediary service agencies, etc. Here, we learned from the statistics of patent growth by Liu Zimei and other scholars [23]. Coefficient contribution, the joint development factor, and the financial input factor were obtained through the expert consultation method, and the number of intellectual property talents and the number of intermediary service agencies were obtained from the statistical yearbook.)

(4) R&D capital input = Input of industry–university–research cooperation * transformation factor of input of industry–university–research cooperation * 0.32+ output of transformation of scientific and technological achievements * input coefficient * 0.25 + 44.56 (The data of input of industry–university–research cooperation came from the internal expenditure of R&D funds in the China Statistical Yearbook of Science and Technology, listed in table function form (see Equation (9)). This Equation was obtained by the regression analysis method);
(5) Enterprise innovation factor investment = "R&D investment" * 0.325+ Enterprise intellectual property special capital investment * 0.511 + 33.51;
(6) Enterprise innovation motivation = policy support * 0.35+ of intellectual property strategy * 0.5+ financing capacity of intellectual property pledge * 0.25;
(7) Innovation revenue = enterprise innovation factor input * 6.5 + 33.38
(Above (4)–(7) Equations were obtained from regression analysis);
(8) Output of transformation of scientific and technological achievements = transaction amount of technology market * transformation rate of scientific and technological achievements = 0.218, which is based on the trend of transaction amount of technology market and ratio of patent application from 2015 to 2020;
(9) Input from industry-university-research cooperation = WITH LOOKUP (time, ([(2015, 1801.23)–(2020, 3002.43)], (2015, 1801.23), (2016, 2026.87), (2017, 2260.06), (2018, 2504.43), (2019, 2779.52), (2020, 3002.43)).

Similarly, the data on the number of intellectual property talents and the number of intermediary service agencies were obtained from the China Statistical Yearbook and regional research, and are listed in the form of table functions. The number of intellectual property talents is taken from the statistical yearbook, including engineering and technical personnel, agricultural technical personnel, and scientific research personnel.

### 4.3. Model Test

In order to understand clearly the driving evolution path of intellectual property rights to regional economic growth, a large system was divided into three subsystems, and the operation path of the three subsystems and their mutual relations were analyzed. Then, the driving and influencing relations among the three subsystems were constructed. The

model equation was established by analyzing the driving force and thrust force of the system dynamics. For the sake of verifying the correctness of the model equation to build, and the causality between driving factors and the correctness of the established regression equation, the model output was compared with the actual system real historical data and the error between the data size was analyzed. To test that the causal relationship between these factors is accurate, the effectiveness of the system dynamics model can be verified. In this paper, the output results of the model were compared with the real historical data of the actual system to see the error between the data to check whether the causal relationship between the above factors was accurate and that the validity of the system dynamics model can be tested. For example, in this study, referring to the research of Liu Zimei and other scholars, the important influencing factors of knowledge stock were decomposed into the number of intellectual property talents, the number of intermediary service agencies, the number of financial investments, the size of the joint development, etc., and the equation was constructed. Knowledge stock was defined as the sum of patent applications and grants, trademark applications, copyright, new plant variety authorization, etc. Intellectual property talents, intermediary service agencies, and financial input were obtained from the statistical yearbook. The fitting showed that the error between the actual knowledge stock data and the statistical data in the intellectual property white paper of Jiangsu Province over the years was between 2.8% and 5%, which confirms the reliability of the system equation model. Similarly, the technical trading volume and enterprise innovation investment were respectively verified. Compared with the historical real data, the error was controlled within 5%, and the fitting degree was between 0.7–0.9, which is significant. It can be seen that the system dynamic model was reasonable and effective on the whole, and it can objectively reflect the correctness of the causal relationship between the driving path and influencing factors of intellectual property management on regional economic growth, which is in line with the actual situation and can be used for further research.

*4.4. Model Simulation and Analysis*

4.4.1. The Data Simulation of Intellectual Property and Technological Innovation in Jiangsu Province

The relevant data of intellectual property and technological innovation in Jiangsu Province were selected to systematically simulate the changes of intellectual property on economic growth from 2015 to 2020. The system simulation was repeatedly compared with the actual situation, and the parameters of the model were adjusted to a reasonable range to obtain relevant values [24]. The simulation results are shown in Figure 2.

- Several key factors were selected to analyze. The analysis showed that regional economy was in the trend of steady growth. This is inseparable from the implementation of the intellectual property strategy since 2009, the implementation of Jiangsu Provincial government's plan to strengthen the province of intellectual property, and the promotion of the intellectual property strategy. According to the above figure, compared with other factors, R&D investment contributes more to regional economic growth, showing a steep slope. This shows that, in the management of intellectual property rights to the mature stage, the continuing growth of R&D spending is still the key element of technology innovation. This has to do with Jiang-Hong Zeng [25] and other scholars that believe that high-tech enterprise R&D investment is the basic way to maintain sustained growth and competitive advantage of the view consistently [26]. In addition, the government's R&D investment is more important. When the government's R&D investment reaches and stays at a certain level, then enterprises are stimulated to increase the R&D investment in such an innovative atmosphere, and the R&D investment needs an abrupt process of a large scale.

- Looking further at the chain structure in the dynamic flow diagram of the system, it mainly comes from the opening of the intellectual property market. With the promotion of the intellectual property strategy, the intellectual property market of Jiangsu Province was active, and the transfer of intellectual property in the technology market

also showed a trend of steady growth. According to statistics, by the end of 2015, there were more than 1800 intellectual property service agencies of all kinds, and 2230 people were qualified as patent agents. The number of trademark agencies reached 685, three times that of the end of the 11th Five-Year Plan. Emerging intellectual property services such as information retrieval, analysis and evaluation, pledge financing, finance and insurance, strategic consulting, and industrial early warning are gradually emerging. At the same time, with the close deepening of industry–university–research cooperation, the transformation of scientific and technological achievements is also strengthened, which makes the government's investment in research and development continuously increase at the macro level, as well as enterprises' investment in innovation factors, and enterprises eventually produce innovative products.

- From the perspective of the promotion of the implementation of the intellectual property strategy and the release of various policies, at the end of the decade of the implementation of the intellectual property strategy and according to the new situation, there were new tasks and new requirements during the 13th Five-Year Plan period. The Jiangsu Provincial government has stepped out of the road of building a leading province with strong intellectual property rights that has the characteristics of Jiangsu and meets the requirements of the times. We will guide innovation subjects to strengthen their awareness of an intellectual property operation and implement the strategic application of intellectual property. We will strengthen the construction of operating institutions, establish market-oriented operation mechanisms, innovate operation models, promote the acquisition, storage, development, portfolio, licensing, transaction, and investment of intellectual property, accelerate the flow and utilization of intellectual property, revitalize intellectual property assets, and accelerate the realization of the market value of intellectual property. In the process of market research and data collection, it was found that, in 2017, Jiangsu Province proposed to promote the construction of a high value patent cultivation demonstration carrier, focusing on strategic emerging industries and characteristic advantages industries, to strengthen the patent intelligence analysis and early warning and patent layout of R&D projects, and to cultivate high value patents. At the same time, in order to solve the shortage of funds, we will support enterprises with intellectual property rights to pledge financing loans, so as to broaden the channels for enterprises, especially small and micro private enterprises and "entrepreneurship and innovation" enterprises to obtain loans, and to promote the easing of financing difficulties.

### 4.4.2. Sensitivity Analysis

- Constant change. By increasing the constant terms by 20% at the same time, it was found that:

Regional gross output value changes greatly. This is mainly caused by the increase of enterprises' innovation motivation and the investment of intellectual property special funds. It can be seen from the following figure that the change of constant term has a great impact on the extraction of indicators. At a certain period of intellectual property development, how to stimulate the motivation of technological innovation of enterprises and promote the special investment of intellectual property are the key issues to overcome to promote regional economic growth. In order to realize a strong chain of high-value patents and strong enterprises, the intellectual property investment of major innovation carriers such as Zijinshan Laboratory, Gusu Laboratory, and Taihu Laboratory should be increased to support their high-value patent output. At the same time, the innovation pilot of financial products such as "intellectual property securitization products", "Billions of financing action", and "intellectual property bond products" are promoted. Under the condition of fully considering the risks of intellectual property investment, the financial policy of the Jiangsu model suitable for intellectual property development is explored, and a perfect investment system is built. Secondly, further research shows that policy support

and intellectual property strategy implementation are not sensitive factors, and the changes of indicators such as knowledge stock and R&D investment are not obvious.

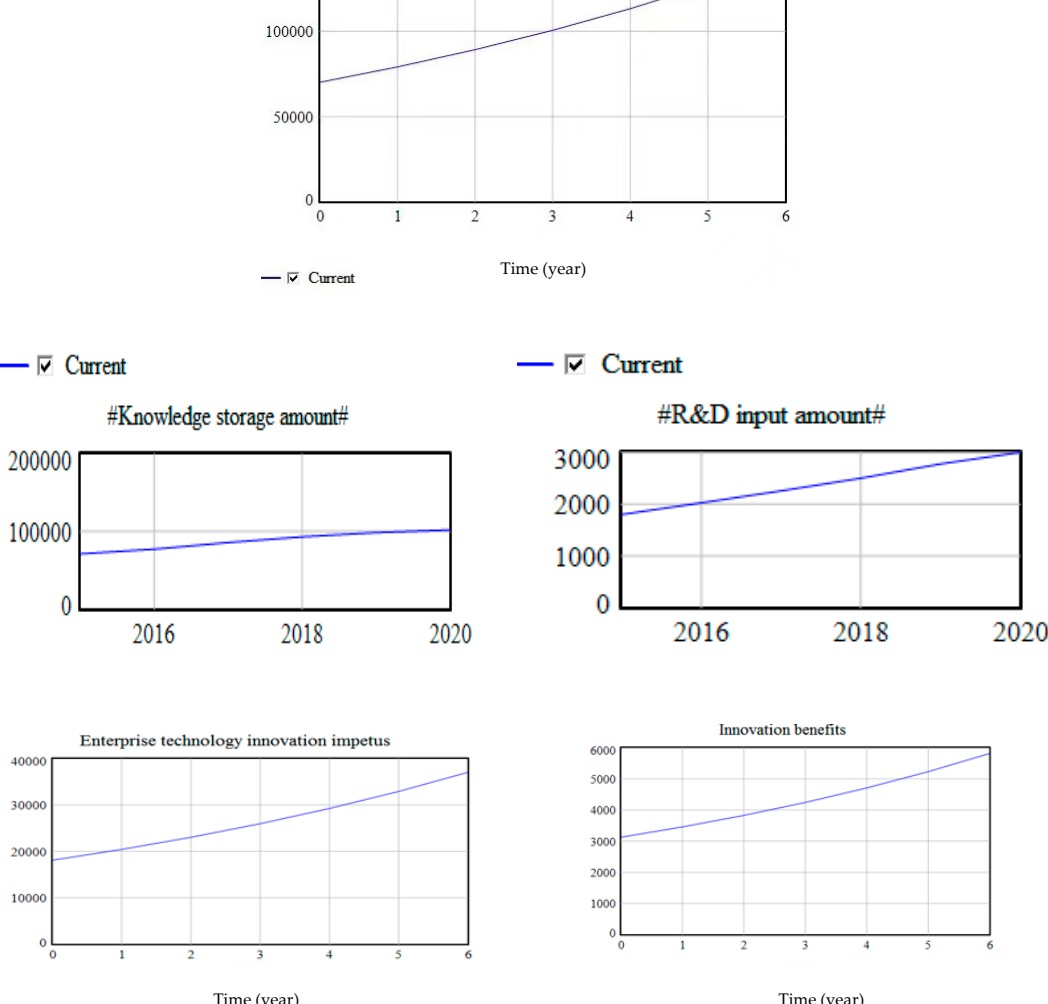

**Figure 2.** Regional gross output value and simulation group diagram of the main factors. (Note: In the above Figure 2, the horizontal axis represents the year, and the vertical axis represents the quantity).

The reasons are as follows. First of all, the economic development of intellectual property rights in Jiangsu province, in the early part of the intellectual property strategy implementation, proposed the strategic measures of a strong province of intellectual property rights and introduced the relevant policies and measures. This was performed to perfect the intellectual property policy system and management system so that they were mature and stable. In normal circumstances, we can conclude that the intellectual property strategy is sensitive elements. Second, in Jiangsu province, the intellectual property strategy implementation of patent application and grants has become the first nationwide. Jiangsu province has established the intellectual property rights trading platform, promoted the sharing of intellectual property, made full use of the intellectual property rights, and can not only give benefits to people with the transferee, but it also can promote the industry related to intellectual property business orderly and with rapid development. From the perspective of the intellectual property market transaction in Jiangsu Province, the frequency and utility of intellectual property transfer are high, indicating that the market development has been mature, which directly affects the activity of the intellectual property

trade market. The order and security of intellectual property transfer directly affect the transaction environment of intellectual property. As shown in Figure 3.

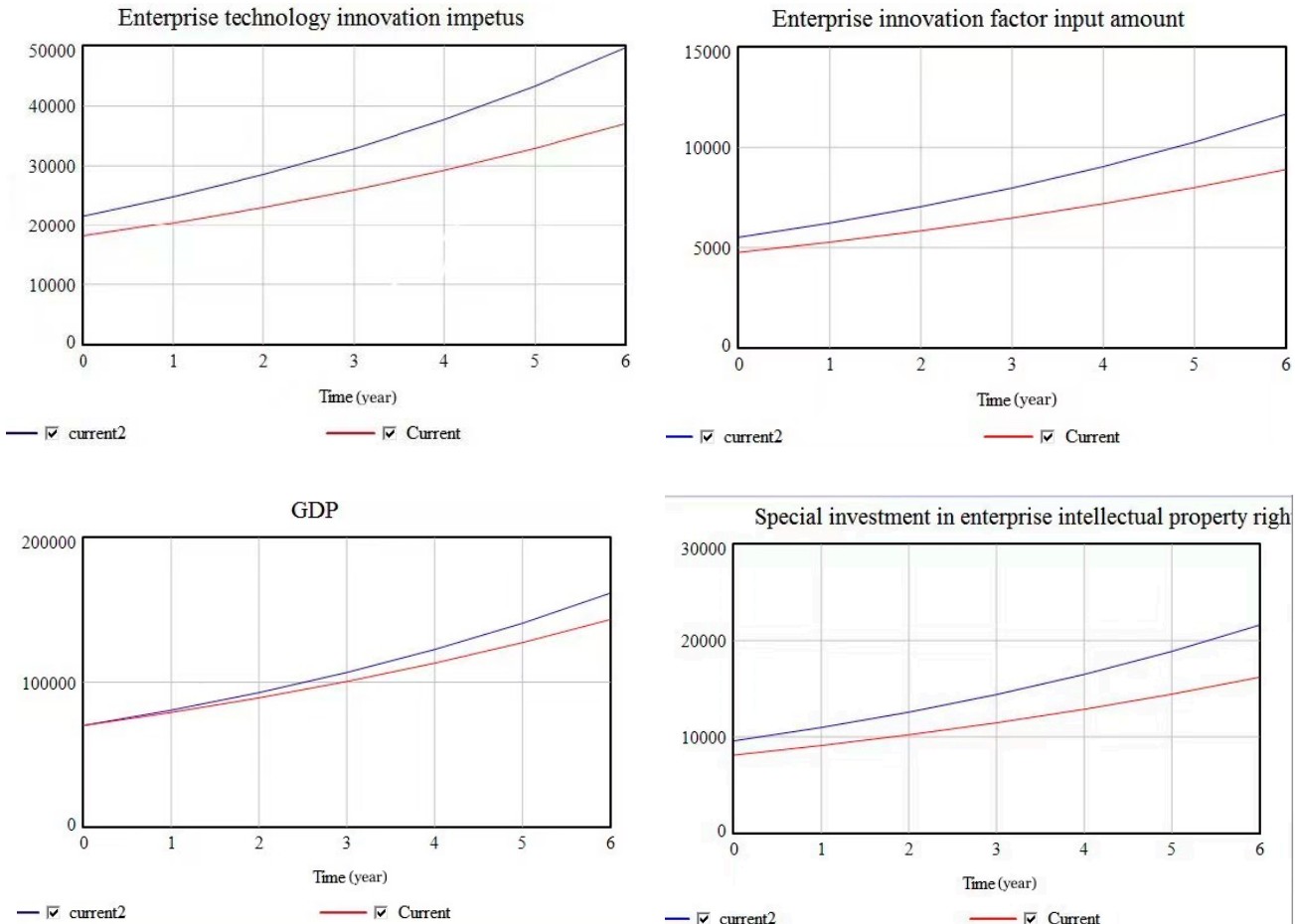

**Figure 3.** Sensitivity analysis of important indexes after constant value increased by 20%. (Note: In the above Figure 2, the horizontal axis represents the year, and the vertical axis represents the quantity).

In addition, as can be seen from the GDP chart, the impact of new policies and strategic layout had a lag effect, which can be seen from the graph with a lag time of 1–2 years, which is consistent with the cycle of public policies from introduction to implementation. Therefore, the administrative department of intellectual property rights should combine the regional economic situation, early layout, and early planning as soon as possible to make decisions in accordance with the local economic development. From the perspective of enterprises, the implementation of policies and the formulation of strategies should be closely coordinated, the development of enterprises should be consistent with the long-term strategic planning of regional development, and decisions should be made as soon as possible to enable enterprises to gain competitiveness. As shown in Figure 4.

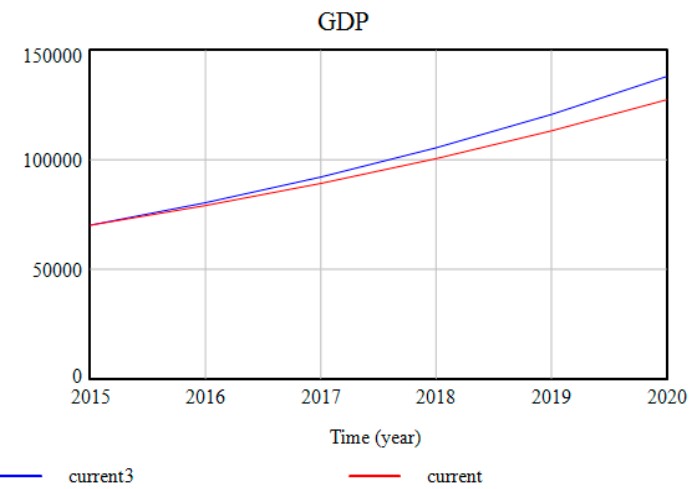

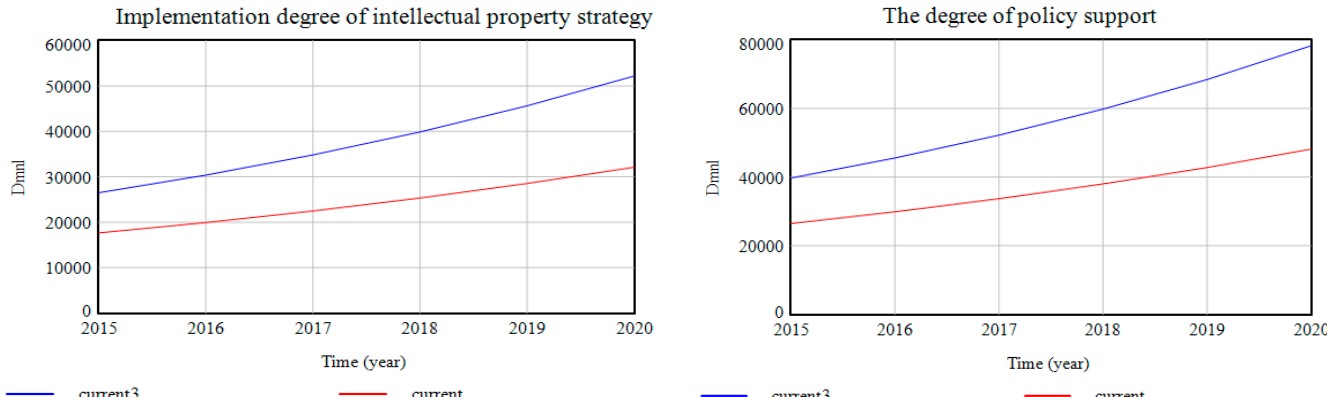

**Figure 4.** The impact of enhanced policy and intellectual property strategy on the regional economy. (Note: In the above Figure 2, the horizontal axis represents the year, and the vertical axis represents the quantity).

## 5. Conclusions and Enlightenment

In this paper, the driving effect of the intellectual property management process on regional economic growth was divided into three subsystems: the driving effect of intellectual property management on technology innovation, the driving effect of technology innovation on regional economic growth, and the driving effect of intellectual property strategy and policy support on regional economic growth. These three systems were not isolated but were connected to each other as a unified whole. Information and resources were constantly exchanged between the systems, and there were output and input processes. Through the analysis of this complex system, the system dynamics model was constructed, parameter estimation and variable equation establishment were carried out, simulation and simulation were carried out, and sensitivity analysis was carried out. The following conclusions were drawn.

### 5.1. In The Simulation Interval of The System, the Important Variables in the Three Subsystems Show an Increasing Trend

As can be seen from the causal analysis diagram, regional output growth in the system of income mainly came from innovation. Analyzing from the innovation income, income depended on the input of R&D and innovation. This conclusion is taken from the simulation diagram of the main index of the slope, the other conclusion is from the sensitivity analysis, and it can also be seen that the input index is a very sensitive factor. The reasons are related to the following two factors:

- To increase investment and promote the improvement of intellectual property macro-management capability. The number of invention patents owned by 10,000 people and the proportion of added value of independent brand enterprises in GDP were included in the modern index system. In order to realize the core index system, the government increased the investment of financial funds and strived to create a new pattern of intellectual property. By the end of 2015, the province had 14 national intellectual property pilot cities, 3 national brand strategy implementation demonstration cities, 3 national copyright demonstration cities, 30 strong counties (districts) with national intellectual property pilot demonstration, 23 national intellectual property pilot demonstration parks, and 4 national copyright demonstration gardens (bases), which are the first in the nation.
- The use of intellectual property rights and the growth of output capacity. The government strengthened intellectual property exchange, established trading platforms and centers, and promoted technology transfer and exchange inside and outside the province and at home and abroad. Through provincial patent technology implementation, transformation of scientific and technological achievements, and guidance of key technological innovation projects and other planned projects, it supported the industrialization of 15,000 major patents, achieving a sales revenue of more than CNY 500 billion. In 2015, the transformation and implementation rate of enterprise patents exceeded 70%, the export value of the copyright industry exceeded USD 30 billion, and the added value of the copyright industry exceeded CNY 540 billion, accounting for more than 8% of the GDP. The added value of the core copyright industry exceeded CNY 300 billion, and the added value of independent brand enterprises accounted for 11% of the GDP. The output value of products with independent intellectual property rights accounted for more than 50% of the total output value of industries above designated size.

*5.2. The Policy Support and The Promotion of Intellectual Property Strategy Are the Strong Guarantee and Drive of Regional Economic Growth*

The market and the government are not one or the other in driving innovation. The increase of the market process and the strengthening of the government institutionalization factor have a significant promoting effect [26]. This is the institutional basis of innovation-driven industrial structure upgrading and technological progress.

*5.3. Intellectual Property Talent Construction, Intermediary Service Platform Construction and Industry-University-Research Cooperation Are Important Poles for Regional Economic Growth*

- To meet the needs of intellectual property development, intellectual property personnel training should catch up with the pace of intellectual property strategy development. They should insist on planning a guide and a demand guide and constantly improve the intellectual property talent evaluation mechanism, further pushing forward the construction of the intellectual property talent cultivation carrier efforts on intellectual property talent training. The construction of a high-quality talent team of intellectual property should be promoted to speed up the lead type of intellectual property rights under the new situation and the strong province construction to provide a solid talent support and assurance.
- To build an innovation-oriented country, we should attach importance to the cultivation of the technology market and the connection between innovation input and industry by technology intermediary, which is of vital importance to promote the construction of a market-oriented innovation system with deep integration of industry, university, and research and to accelerate the industrialization of innovation achievements. It is necessary to improve the ability of intellectual property innovation and application, improve the intermediary service system of intellectual property, integrate information resources on the basis of various technology innovation and service platforms, build innovation information platforms of industry colleges, and provide comprehensive information on technology research and development, public service, science and technology resources, and other kinds of information. Using the national intellectual property operation and

public service platform, the trading operation pilot platform in the patent transfer into buy-hosting, trade-flow. It can also to use the platform to carry out pledge financing, such as increasing the benefit of the patent application, in order to solve the intellectual property rights achievements conversion rate, low level of marketization, and intellectual property right transfer channels, financing difficult problems such as intellectual property.

● The industry–university–institute cooperation is made of enterprises, universities, and research institutes used to achieve a way to effectively integrate high quality resources and to form a strong research, development, and production integration of the system. A production–study–research cooperation can promote the increase of the number of intellectual property rights on the one hand, and, on the other hand, it can promote technological achievements of property rights, industrialization, and capitalization and can speed up the transformation of scientific and technological achievements to real productivity. The government establishes the policy framework to provide personnel to enterprises and research institutes.

*5.4. Enterprise Technological Innovation Is the Carrier of Intellectual Property Management Activities and the Necessary Path for Regional Economic Growth*

Enterprises continuous use of fiscal support and their own funds to obtain competitive. Accompanying by a political alliance between activities, continuous innovation, under the guidance and the support of regional policy, enterprises promote the upgrading of industrial products, even to the original industrial chain in the field of new breakthrough innovation activities, the end result for high value of patents and innovative products. Enterprises obtain a fat profit, accordingly, regional economies have grown.

## 6. Future Research Direction

This study paid more attention to the interaction and cooperation between the government, enterprises, and research institutions, as well as the complete performance track and function process of intellectual property management activities in the whole activity. In the management activities, the introduction of the system dynamics method, from the perspective of the system, matches the main body and management activities to simulate the effect of the paths and display technology innovation activities in the management activities and the mediation relationship between economic growth, breaking through the intellectual property management activities of quantitative research on economic growth.

This study selected samples with strong comprehensive strength of intellectual property management for research, and future research is expected to focus on the following aspects:

(1) The function and feedback of the intellectual property management system are directional, and the function and driving force are also different. Are there differences in different samples?
(2) With the help of the system dynamics method, the role process of intellectual property management activities on economic growth and the cooperative relationship between subjects are studied. Are the paths explored by other methods, such as the fuzzy set qualitative analysis method, consistent with the paths analyzed in this study?

**Author Contributions:** Conceptualization, X.Y. and Y.Q.; methodology, Y.Q.; software, X.Y.; validation, X.Y.; formal analysis, X.Y. and Y.Q.; investigation, X.Y. and Y.Q.; resources, X.Y.; data curation, X.Y.; writing—original draft preparation, X.Y.; writing—review and editing, X.Y.; visualization, X.Y.; supervision, Y.Q.; project administration, Y.Q.; funding acquisition, X.Y. All authors have read and agreed to the published version of the manuscript.

**Funding:** This research was funded by the National Social Science Foundation of China, grant number 16BZZ074.

**Informed Consent Statement:** Informed consent was obtained from all subjects involved in the study.

**Data Availability Statement:** The data of variables can be obtained in three ways: (1) directly from the statistical yearbook, such as the initial value of regional gross output value, the number of intellectual property talents, the number of intermediary service agencies, and the input of industry–university–research cooperation; (2) using regression analysis, such as knowledge stock, R&D investment, etc.; (3) using the acquisition of expert consultation law, such as the intensity of joint development, the implementation of intellectual property strategy, and the intensity of policy support, which will be valued as 1 and given weight.

**Conflicts of Interest:** The authors declare that they have no conflicts of interest to report regarding the present study.

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
