# Peer review of "Simulation of Intellectual Property Management on Evolution Driving of Regional Economic Growth"

_applsci, doi:10.3390/app12189011_

Round 1
Reviewer 1 Report
Thank you for giving me the opportunity to review the article entitled " Simulation of intellectual property management on evolution driving of regional economic growth".
I found it a very interesting topic and one on which there is a lot of research to be done. However, the article needs to undergo some modifications in order to be published.
In the introduction many aspects are missing, such as: GAP to be covered, main contributions, sample on which the research is carried out, research questions to be solved. As well as a final paragraph showing the structure of the rest of the article.
There are some flaws in the wording of the article, as for example in line 183 that starts with lowercase letters.
I miss the establishment of hypotheses.
Regarding the literature review and citations in general, I consider that a research article for a journal of this level cannot contain only 24 citations. The authors have to do more research work.
I think it would be better to make an adequate separation between methodology and results. Also, a discussion section is necessary in which the results are contrasted with others contributed by previous researchers.
In the conclusions, the authors should point out the main contributions and applications of their research. It is also necessary to show the limitations to the scope of the study and, if applicable, the possible lines of future research.
Until the above has been implemented, I consider that the article cannot be published.
Reviewer 2 Report
The authors presented the results of a study of the relationship between GDP and intellectual property. System dynamics is used as a research method.
On the one hand, the obtained results and conclusions look interesting and important, but on the other hand, they are obvious and insufficiently substantiated.
I propose the following improvements to the manuscript to enhance the scientific value of the study.
1. The literary review is limited and does not give the reader the necessary idea of ​​the state-of-the-art. First, I propose to present a more detailed analysis of known studies for causal. At the same time, this analysis will be the substantiation for author's causal loops, as well as the material in section 3.1.2. Secondly, after all, GDP growth depends mainly on the implementation of real projects in the economy, in particular, infrastructure development projects. Intellectual property management has an indirect impact on this main factor of economic growth. Thus, I propose to present an analysis of studies on the interaction of large infrastructure projects and GDP. For example, in DOI: 10.1007/978-3-319-78295-9_1, the One Belt and One Road initiative is being studied as such an infrastructure project, etc.
2. The conclusions are not explicitly confirmed by the simulation results. I propose to move specific results to section 4.4 and detail how they are obtained by model.
3. The proof of the reliability of the constructed model is not obvious. I propose to show on the graphs of Fig. 2 and possibly in tabular form the difference between model and actual data.
4. I propose to specify the data sources that were used to calculate the coefficients of the regression equations in Section 4.2.
5. I suggest presenting the structure of the research process in graphical form, such as a flowchart. This will increase reader interest in the article.
A few minor comments and suggestions.
6. Paragraphs are very long, which makes it difficult to understand the author's ideas and thoughts. I propose to separate individual thoughts into appropriate paragraphs.
7. The text in lines 321 - 329 is clearly improper and refers to the previous paragraph.
8. The legend on the graphs needs to be explained. For example, it's not clear what "current2" and "current3" are. On the other hand, there is no need for "current" in Figure 2.
9. The axes of all charts must be labelled.
10. All abbreviations, except for known ones, should have explanations (for example, in section 2.1).
11. The reference is missing on line 508.
Reviewer 3 Report
The paper is well theoretically grounded and the model presented has been estimated correctly. The article has an important scientific contribution considering the limitations of its application. There are some formatting (graphics and model) and referencing details that may be corrected in the final version to be submitted. Format should be improve and adapted to the journal style.
Reviewer 4 Report
In my opinion the article „Simulation of intellectual property management on evolution driving of regional economic growth” is very interesting and worth publishing. However, I have some proposals to improve it.
1) I would suggest revising the title. The word “simulation” used in the title seems to be not appropriate. I would use “intellectual property rights management”. The phrase “on evolution” is unclear. What kind of evolution do You mean?
2) The phrase “intellectual property rights” ought to be use consequently within the whole article (not only intellectual property).
3) In the abstract the authors used the expression “promotion of intellectual property strategy”. The reader might not know whose strategy You mean. It should be clearly defined.
4) In the Introduction section there are no references in lines 26-30.
5) The authors write that “at present, the speed of economic development is slowing down”. They should define clearly whether they think of: the global economy, a chosen country or a region. They should also point out precisely what period of time they mean.
6) In lines 35-36 the word “upgrading” is used twice.
7) Referring to lines 36-39 I might say that this sentence is a kind of slogan because there are many more factors that contribute to market competitiveness and not only the improvement of the management level of intellectual property rights.
8) In line 67 the authors use the word “digitization”. Shouldn’t it be digitalization?
9) In line 81 the authors use the phrase “green intellectual property”. It needs some explanation.
10) In lines 84-85 there is a sentence taken out of context “Promote the standardization of intellectual property by settings goals and strategies in the field of intellectual property”. This sentence should be rephrased.
11) When abbreviations are used the authors should develop them first and then can use them in the abbreviated form, for example, IPR, FDI, and others.
12) The article should be written in the impersonal form and in the appropriate style (see: line 176).
13) In line 183 start the sentence with the capital letter.
14) In the third part of the article the authors have divided the levels of factors into macro and micro. This choice should be justified because they have not taken into consideration the mezzo level.
15) In fig. 1 it is written “growth anount” but should be “amount”. Please correct it.
16) I suggest modifying the title of fig. 1 into “System dynamics flow of IPR influence on regional economic development”.
17) The choice of Jiangsu Province for the research should be clearly justified by the authors and they ought to explain what the limits of the research are.
Reviewer 5 Report
The presented research has the potential for publication in Applied Sciences. Below are the points that should be taken into account before adopting an article:
Abstract and keywords: I recommend adding a research goal to the abstract. It would be better not to mix keywords with the title of the manuscript.
Introduction: the introduction is very general and does not follow the formulated goal (only in the broad sense of the word). I consider it necessary to indicate at the outset what the knowledge gaps are in the context of the mentioned topic and how the research will contribute to filling this gap.
Literature review: in the context of such current topics, I consider it necessary to cite the literature of impacted journals up to 5 years old. Alternatively, I recommend combining the individual statements of the authors with other research. In fact, the article does not contain any literature review. I would also like to specify the hypotheses.
The quality of the manuscript would benefit from the addition of Methodology.
Conclusion: please state the findings, especially in the context of the formulated goal (which appears in the article only once and is not further developed in any way). I believe that the research is of good quality. The quality of research would be enhanced if the results were clearly structured.
Round 2
Reviewer 1 Report
Although the authors have fixed some of the rulings I indicated to them, the most important ones follow. For this reason, I deny its publication.
Author Response
Thank you for the reviews.

Reviewer 2 Report
Unfortunately, the authors considered only some minor suggestions. Regarding my principled comments, I agree with the authors' response to the first point. The authors ignored the other proposals.
I do not consider the article in this form useful for readers and cannot recommend it for publication.
Author Response
Thank you for the reviews.

Round 3
Reviewer 1 Report
Now, the paper is correct
Author Response
Thank you for your reviews!
Reviewer 2 Report
The authors considered most of my suggestions in the new manuscript version. However, I am compelled to repeat my two proposals.
1. Please add references to data sources (including Internet resources) that you used to calculate the coefficients of the regression equations. In this case, readers and future researchers will be able to repeat or continue the study.
2. Please provide in the text the decoding of the designations “Current”, “current2” and “current3” (Figures 2 – 4).
